# MT-GBM: A Multi-Task Gradient Boosting Machine with Shared Decision Trees

## Abstract

Despite the success of deep learning in computer vision and natural language processing, Gradient Boosted Decision Tree (GBDT) is yet one of the most powerful tools for applications with tabular data such as e-commerce and FinTech. However, applying GBDT to multi-task learning is still a challenge. Unlike deep models that can jointly learn a shared latent representation across multiple tasks, GBDT can hardly learn a shared tree structure.

In this paper, we propose **M**ulti-**t**ask **G**radient **B**oosting **M**achine (MT-GBM), a GBDT-based method for multi-task learning. The MT-GBM can find the shared tree structures and split branches according to multi-task losses. First, it assigns multiple outputs to each leaf node. Next, it computes the gradient corresponding to each output (task). Then, we also propose an algorithm to combine the gradients of all tasks and update the tree. Finally, we apply MT-GBM to LightGBM. Experiments show that our MT-GBM improves the performance of the main task significantly, which means the proposed MT-GBM is efficient and effective.

## 1 Introduction

Gradient boosting is a powerful machine-learning technique that achieves state-of-the-art results in a variety of applications. It is a process of constructing a strong predictor by ensembling weak predictors and performing gradient descent in a functional space. Gradient boosting has achieved tremendous success in a wide range of machine learning tasks. It is popular in applications with heterogeneous features, noisy tabular data, and complex dependencies, such as recommender systems (Wang et al., 2016; Wen et al., 2019), web search (Agrawal & Shanahan, 2010), FinTech (Wang et al., 2018; Mei & Li-ren, 2015; Yu-Zhuo & Xiao-Ni, 2006), online advertising, etc (Roe et al., 2004; Zhang & Haghani). Among all gradient boosting algorithms, XGBoost and LightGBM are the two most famous algorithms. They have contributed to countless top solutions in machine learning and data mining competitions on tabular data.

Despite the considerable success of gradient boosting, applying GBDT models to multi-task learning is still challenging. The GBDT model aims to train a high-performance discriminative model towards a single loss function of a single task. Therefore, it cannot learn a shared latent feature representation, which is essential for multi-task learning.

We list three things that limit the applicability of GBDT in multi-task learning.

First and foremost, the core of efficient multi-task learning is to extract valuable hidden representation that shares across all tasks, which is non-trivial for tree-based ensemble learning methods like GBDT. For example, in news recommendation, user response prediction tasks include the prediction of click-through rate, dwell-time, thumbs, reviews, shares, and so on. These objects correspond to the same user; thus an informative representation of the user is the key to such a multi-task learning problem. The recent development of deep learning can achieve this naturally using hidden embeddings and layers in neural networks (Ruder, 2017; Huang et al., 2018; Gao et al., 2019), thus has shown its dominance for multi-task learning in computer vision and natural language processing (Su et al., 2015; Radford et al., 2019; Liu et al., 2019). However, it is known that neural networks sometimes fail in applications with tabular data.

Second, the splitting mechanism for decision trees aims to separate the input space as well as the training samples according to a single loss function, which could not be applied directly to multi-

task learning where there are multiple losses. When optimizing end-to-end neural networks, it is natural to sum up the gradients from all losses. However, for tree-based models, the splits that have maximal information gain for the losses can differ a lot, especially when training classification and regression tasks at the same time. Therefore, it is challenging to find tree splits that are suitable for all tasks.

Third, in multi-task learning, the model should keep similar paces for each task for stable learning. For neural networks, current gradient descent methods with adaptive step size can adjust the learning rate for different tasks. However, the learning speed of GBDT is related to the residual errors. Therefore, when there are multiple tasks, if a fixed learning rate is applied to all tasks, the model tends to overfit certain tasks while underfit the others. Therefore, how to adjust learning rates for multiple tasks remains a challenge.

In this paper, we propose a new algorithmic technique for multi-task learning based on GBDT, namely multi-task Gradient Boosting Machine (MT-GBM). The key novelty of MT-GBM is that it learns shared decision tree structures for multiple tasks, which can be understood as the hidden knowledge across all tasks. As the decision trees are shared, it requires almost the same computational overhead comparing to a single task GBDT. Therefore, MT-GBM can be efficiently trained. We verified MT-GBM on two financial datasets. Experimental results show that MT-GBM can achieve significantly higher performance than existing state-of-the-art gradient boosted decision trees methods, XGBoost (Roe et al., 2004), LightGBM (Ke et al., 2017) and Catboost (Prokhorenkova et al., 2018) .

Our contribution can be summarized as follows:

- We propose MT-GBM that learns an ensemble of shared decision trees learned from multiple losses simultaneously, which significantly improves the efficiency and effectiveness of GBDTs for multi-task learning.
- We propose a new learning algorithm that adaptively adjusts the learning rate for each task by taking the scale of residual errors into consideration. It stabilizes training on arbitrarily complex problems.
- Our method is easy to implement upon existing GBDT models such as XGBoost and Light-GBM. We tested our method by modifying LightGBM and showed significant performance improvements.

## 2 PRELIMINARIES

### 2.1 MULTI-TASK LEARNING WITH DECISION TREES

It is widely known that multi-task learning could improve the performance of every single task by sharing knowledge from all tasks. When using max-margin machine learning (i.e., SVMs) or deep learning (i.e., feed-forward neural networks), multi-task learning has achieved significant success (Caruana, 1997; Evgeniou & Pontil, 2004; Bingel & Søgaard, 2017). However, with decision-tree-based models, it remains unclear how to learn shared knowledge from multiple tasks.

There are many potential reasons why adding extra outputs to a model might improve generalization performance. For example, to the extent that tasks are uncorrelated, their contribution to the aggregated gradient, which guides the node splitting in GBDT, can appear as noise to other tasks. Thus uncorrelated tasks might improve generalization by acting as a source of the noise. Figure 1 (a) shows a situation of data samples with a noisy label. One single sample contains two rectangles while the left represents label 1 and the right represents label 2. We can assume that task 1 clicks behavior and task 2 is shares behavior in the recommendation system. It makes it clear that two tasks fix the splitting line from dotted line to solid line. Figure 1 (b) shows that Sub tasks also help the sample splitting when dataset has the class imbalance problem.

### 2.2 PROBLEM DEFINITION

GBDT is an ensemble model of decision trees, which are trained in sequence. In each iteration, GBDT learns a new decision tree by minimizing the negative gradients (also known as the residual

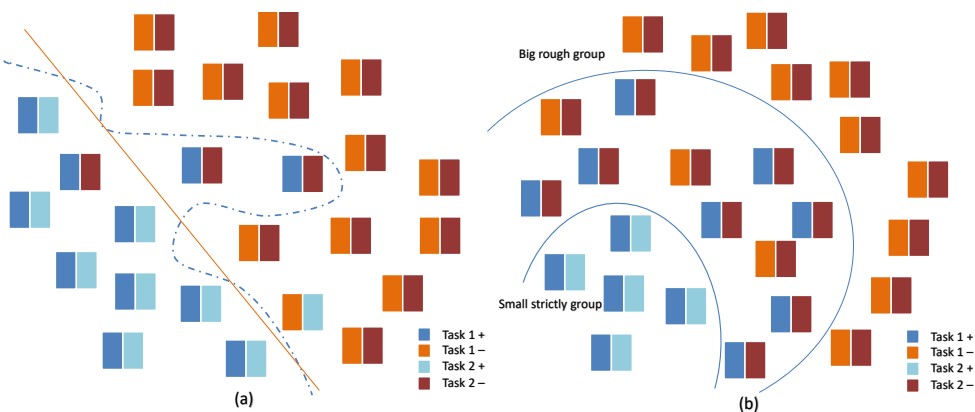

Figure 1: (a) Samples with noisy tasks. Each sample contains two rectangles as tasks and the color means whether it is positive or not. There are some noises in each task in the middle of the figure. Take two tasks into consideration is better than any one of them;(b) Samples with sub tasks. Task 2 is a subclass of task 1, and it helps the model to split the small group from the big group which is not accurate enough. Finally model will predict a high probability for sample in small group and a medium in big group.

errors) of previously trained trees. Specifically, to construct the decision trees, the algorithm finds the split points in the feature space according to the residual errors.

Assume we have $n$ learning tasks $t \in \{1, 2, ..., n\}$, each with a set of supervised data $\{(x_1, y_1), ..., (x_m, y_m)\}$. For simplicity, we assume that the number of data points $m$ in each task is the same, and all the tasks share the same input space. For the $t$-th task, there is a predefined learning objective $L_t$. In regression, we might consider the mean squared error:

$$L_t = \frac{1}{m} \sum_{i=1}^{m} (y_i - p_i)^2 \tag{1}$$

And we use the negative log-likelihood as the classification loss:

$$L_t = -\frac{1}{m} \sum_{i=1}^{m} (y_i \log p_i + (1 - y_i) \log(1 - p_i)) \tag{2}$$

Finally, we assume the main task objective function $L_1$ as the metric to evaluate the model.

## 2.3 RELATED WORK

In this section, we ignore the related work of neural network because most of neural network work well in specific tabular dataset (Zhang & Yang, 2018; Ruder, 2017) .

To achieve a multi-task tree, there were some works about multi-classifier tree on Adaboost (Faddoul et al., 2012) . This paper revises the information gain rule for learning decision trees and modified MT-Adaboost to combine multi-task Decision Trees as weak learners. Since its algorithm depends on their new equation of Information Gain, the model does not support doing the regression task yet. Obviously, it fails when it comes to residual errors in GBDT cause the loss is float number even in classifier tasks.

Another approach to multi-task learning is the work sharing tree structure for different tasks (Chapelle et al., 2011). This paper presents a general solution of boosting for different task types by minimizing the sum of losses $\sum_{i=1}^{n} L_i$. It uses a relative value to control the strength of the connection between the tasks. We approve of the way about exploring the general solution. But the part of optimization in which splitting the node would be hard and slow when consider minimizing all loss functions. It seems to be a bit complex and obsolete and is beaten by the existing implementations of gradient boosted decision trees such as XGBoost and LightGBM.

However, the GBDTs has some tricks to deal with multi-class task (Fan et al., 2017), i.e. training $n$ trees in one iteration for $n$ classes. When inferring, the trees belong to the specified class will be activated.

---

**Algorithm 1** Traditional Multi-Class GBDT

---

**Input:** $n$ the number of tasks, $m$ the number of samples. Accumulative prediction of previous rounds:$\{p_{1,t}, \ldots, p_{m,t}\}_{t=1}^n$; Goundtruth label:$\{y_{1,t}, \ldots, y_{m,t}\}_{t=1}^n$
**Output:** New decision trees $T_1, T_2, ..., T_n$ in this iteration.

1: **for** $i = 1, \ldots, m$ **do**
2:     $p_{i,t} = \frac{\exp(p_{i,t})}{\sum_{t=1}^n \exp(p_{i,t})}$, $t = 1, \ldots, n$
3:     **for** $t = 1, \ldots, n$ **do**
4:         // J is the number of nodes in the tree
5:         // N is the number of samples in the node j
6:         $\tilde{y}_{j,t} = y_{j,t} - p_{j,t}$, $j = 1, \ldots, N$
7:         // $R_{jti}$ means that the $i$th sample belongs to the $j$ node at the $t$th task
8:         // $\gamma_{jti}$ means the prediction of $i$ sample at the $t$th task
9:         $\{R_{jti}\}_{j=1}^J = J - \text{terminalNodeTree}(\{\tilde{y_{lt}}, x_l\}_l^N)$
10:       $\gamma_{jti} = \frac{K-1}{K} \frac{\sum \tilde{y}_{lt}}{\sum |\tilde{y}_{lt}|(1-|\tilde{y}_{lt}|)}$, $j = 1, \ldots, J$
11:     **end for**
12: **end for**

---

But the trees for different classes have no interaction when training except the last softmax transformation. The trees for the label with few positive samples would have a terrible training, since it can not learn things from the other labels.

Moreover, it has a big problem that it costs $n$ times of resources when training and inferring. There were papers that implied the multi-class by solving the sparse outputs problem (Si et al., 2017) and achieving an order of magnitude improvements in model size and prediction time over existing methods. But it still doesn't seem to be a multi-task solution, while realistic datasets above always have dense outputs and multi-class which can be positive at the same time. Therefore we call it a multi-class solution, not multi-task.

To address the limitations of previous works, we propose two new novel techniques called multi-output tree and multi-task splitting.

## 3   Multi-Output tree

The multi-task demands a new structure called multi-output tree to save the intermediate results. The new model should manage to specify weights for each label, and update leaf outputs with specified rates, which make it necessary to multiply gradients of the samples with the number of labels.

In this section, we propose a new structure tree by modifying the traditional decision tree, which almost has no disruptive innovation to the GBDT training procedure. Multi-output Tree shares same structure and same leaf instead of sharing other parameters. And we enrich the node and leaf information which contains Gradients, Hessians, mean value of the data at the node and etc. There will be a discussion about them below:

Firstly, each sample in the dataset should have $n$ predictions, $n$ Gradients and $n$ Hessians, where the $n$ is the number of labels. They should be calculated at the beginning of each iteration like old GBDT. And it causes a problem that the histogram object(Ke et al., 2017) which is used to speed up splitting could only be calculated by one Gradient and one Hessian for each sample.

Then, we separate the concept of Gradients and Hessians. When training the tree, old Gradients and Hessians were designed both for splitting the node and updating the output. In MT-GBM, we denote the Gradients and Hessians used for updating as $G_u$ and $H_u$, respectively. The Gradients and Hessians used for splitting the node are called ensemble $G_e$ and $H_e$, while it has one value for each sample, which determines how the tree splits. And more discussion about the ensemble $G_e$ & $H_e$ and updating $G_u$ & $H_u$ will be shown in the next sections.

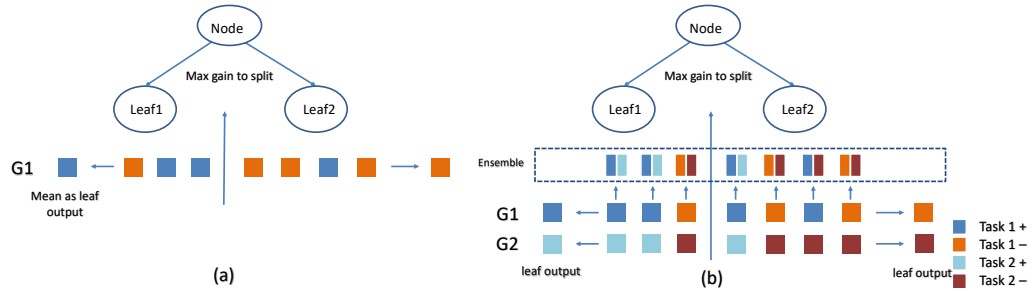

Figure 2: (a) Old structure of gradients in the GBDTs. Each rectangle is the $G$ of one sample. Assume the $H$ equals constant in the square loss and model splits samples by the $G$; (b) New structure of gradients in the MT-GBM. The arrows in horizontal direction mean the transformation generating updating $G$ and the arrows in vertical direction mean the transformation generating ensemble $G$.

At last, we only need to save the mean of residual errors of all the data at each leaf for each label, which was used to calculated residual errors in iterations below. It's easy to support the model to save part of trees for every single label. Therefore the dump model file will be the same size as the traditional GBDT and single label can be inferred as the same speed.

## 4 MULTI-TASK SPLITTING

In this section, we define how to calculate the two types of $G$ and $H$ and their usages in MT-GBM.

XGBoost and LightGBM have done a lot of work to make the GBDT efficient by proposing new loss function and parallel computing when splitting. Their loss functions were based on Gradients and Hessians of the data:

$$L^* = -\frac{1}{2} \sum_{j=1}^{T} \frac{G_j^2}{H_j + \lambda} + \gamma T \tag{3}$$

The loss serves as the most principle of the splitting selection, although XGBoost and LightGBM both have some techniques modify or simplify them in computational procedure. At each leafs, the $H$ and $G$ means the sum of the Gradients and Hessians of all the data on it.

### 4.1 GRADIENTS FOR SPLITTING

The tree splits depending on the loss function, while the sum of two leafs' losses should be minimum at the split position. It would be a large amount of computation if model try to minimize all the losses. Therefore we propose a new Ensemble Gradient $G_e$ and a new Ensemble Hessian $H_e$ in the new equation:

$$L^* = -\frac{1}{2} \sum_{j=1}^{T} \frac{G_{ej}^2}{H_{ej} + \lambda} + \gamma T \tag{4}$$

To consider all the $G$ and $H$ of all type of labels, there would be two functions that calculate the $G_e$ and $H_e$: $G_e = f_G(G_1, G_2, G_3, ..., G_n)$, $H_e = f_H(H_1, H_2, H_3, ..., H_n)$.

Different tasks may bring different distributions of gradients to model. Set a normalization factor $w$ to normalize all the losses of tasks would be beneficial to the ensemble. We normalize them to a distribution with 0.05 mean and 0.01 standard deviation.

We realizes the $f_G$ and $f_H$ have to be designed to outstand only one type of label or several relative labels with positive correlation in one iteration by trials. That is because different labels will be learned or fitted at different rates and the $G$ and $H$ will have a completely different order of magnitude, which is hard to ensemble. Thus we define the $f_G$ end $f_H$ by sum them with weights

---

**Algorithm 2** The calculations of $G_e$ and $H_e$ for one sample before an iteration

---

**Input:** predication of trees above of each task:$p_1,p_2,p_3,...,p_n$;true label of each task:$l_1,l_2,l_3,...,l_n$

**Output:** $G_e,H_e$

1: $n$ = number of tasks
2: **for** $i = 1, \ldots, n$ **do**
3:    $G_i = \frac{d}{dp} L(l_i, p_i)$
4:    $H_i = \frac{d^2}{dp^2} L(l_i, p_i)$
5: **end for**
6: Randomly initialize $\mathrm{w}_1, \mathrm{w}_2, \mathrm{w}_3, ..., \mathrm{w}_n$ to make $\mathrm{w}_i * G_i$ with 0.05 mean and standard deviation 0.01
7: Randomly choice one or more indexes of tasks from 1 to $n$ as $k_1$ to $k_m$
8: **for** $k = k_1 , \ldots, k_m$ **do**
9:    $w_k = \gamma \mathrm{w}_k$, which $\gamma$ is a parameter from 10 to 100
10: **end for**
11: $G_e = \sum_{j=1}^{n}(\mathrm{w}_j \cdot G_j)$
12: Randomly initialize $\mathrm{v}_1, \mathrm{v}_2, \mathrm{v}_3, ..., \mathrm{v}_n$ to make $v_i \cdot H_i$ with 1.0 mean and standard deviation 0.1
13: $H_e = \sum_{j=1}^{n}(\mathrm{v}_j \cdot H_j)$
14: **return** $G_e, H_e$

---

which will randomly concentrate on one of few tasks. The probability of each task be chosen can be assigned which represents the importance of tasks.

The parameter $\gamma$ is used as a factor to outstanding the chosen tasks. It could be verified from 10 to 100 by experiments from case by case. The smaller correlation, which means that there are strong conflicts between tasks, the better result with big $\gamma$. From the algorithm we could find it is complex to keep different $G$ and $H$ ensemble balance. We propose a common one above, and believe there might be other better ensemble algorithms for particular problems and tasks.

When it comes to a problem with main task and sub tasks, $k_1$ to $k_m$ should not be selected randomly. We suggest that to always choice the main task label as the $k$. And there is a helpful method to improve the learning process, which is to calculate or evaluate the correlation of the main task with others and give a bigger w and smaller v for the task of positive correlation.

### 4.2 GRADIENTS FOR UPDATING LEAF VALUE

After defined the $G_e$ and $H_e$, the GBDT tree learner could build trees for each iteration. The length of $G_e$ and $H_e$ should be same as the length of the number of samples, while the length of Updating Gradient $G_{un}$ and Updating Hessian $H_{un}$ same as the length of the number of samples multiplying the number of labels.

It is acceptable for the MT-GBM to set the same rate for each label with traditional GBDT. However, we still have some explorations about the $G_{un}$ and $H_{un}$, and propose the algorithm below with some tricks.

The function $f$ we defined is used to calculate correlation of the gradients vector of each label in current iteration, which means the updating speed should be fast if all the gradients have the similar distribution. Actually whether the calculation of correlation works or not depends on many things such as type of tasks and meaning of tasks. Therefore set the $corr$ as a constant equal 1.0 might be better on some complex datasets, which means the updating rate is same as the traditional single task GBDT. And noticed that we multiply the same $corr$ for each $G_{ui}$ in one iteration, it is because that the GBDT learns things from residual errors. Trees get inconsistent paces if we set different rates for different tasks, which leads to bad results.

## 5 EXPERIMENTS

### 5.1 EXPERIMENT SETUP

Through experiments, we seek to answer the following questions:

---

**Algorithm 3** The calculations of $G_{un}$ and $H_{un}$ for one sample before an iteration

---

**Input:** predication of trees above of each task :$p_1,p_2,p_3,...,p_n$;true label of each task:$l_1,l_2,l_3,...,l_n$
**Output:** $G_{u1},G_{u2},G_{u3},...,G_{un},H_{u1},H_{u2},H_{u3},...,H_{un}$

1: $n$ = number of tasks
2: **for** $i = 1,\ldots,n$ **do**
3:    $G_{ui} = \frac{d}{dp}L(l_i,p_i)$
4:    $H_{ui} = \frac{d^2}{dp^2}L(l_i,p_i)$
5:    concatenate the $G_{ui}$ of all the samples as vector $G_{sui}$
6: **end for**
7: calculate the correlation $corr$ with $n$ vectors $(G_{su1},G_{su2},G_{su3},...,G_{sun})$:    $corr = f(G_{su1},G_{su2},G_{su3},...,G_{sun})$
8: **for** $i = 1,\ldots,n$ **do**
9:    $G_{ui}* = clip(corr,0.5,1)$
10: **end for**
11: **return** $G_{u1},G_{u2},G_{u3},...,G_{un},H_{u1},H_{u2},H_{u3},...,H_{un}$

---

1. Can MT-GBM achieve better performance state-of-the-art implementations of gradient boosted decision trees (GBDTs) XGBoost, LightGBM and Catboost?

2. Does neural networks of multi-task achieves same performance as MT-GBM?

In this section, we report the experimental results regarding our proposed MT-GBM algorithm which was developed on the LightGBM(The code[1]https://github.com/Microsoft/LightGBM is available at GitHub). Particularly, the details of experimental setup, including data description, baseline models, and some specific experiments settings, will be presented below. All experiments are done with the same features in GBDTs, Neural Network, and MT-GBM, and the sub tasks in MT-GBM are derived from the main task. Our experimental environment is a Linux server with two i9-7980 CPUs (in total 36 cores) and 128GB memories.

## 5.2 DATASETS AND BASELINES

We tested the methods on two different real-world datasets. The first one is part of China Foreign Currency Volume from 2017 to 2019. The dataset is about the currency exchange volume between one country and China by common people. The target is simple: use the historic data to predict next day volume and we make the last 30 days value as the testset. It is a time-series problem, and it only contains one amount in every workday from 2017 to 2019 and has no features. Rmse and mape are used to evaluate the models.

GBDT is proved as a great model in time-series cases and features are often be constructed by sliding window (Lee et al., 2018; Liu et al., 2018; Xia & Chen, 2017). We count the minimum, maximum, average, and variance of the last three days, one week, two weeks, and one month as the sliding window features. With these, a baseline is finished and we run XGBoost and LightGBM on it.

And the second dataset is Kaggle Competition:IEEE-CIS Fraud Detection ([2]), which is a credit fraud detection case from Vesta Corporation. The target of model is to predict whether this transaction is fraud or not. There is about 3.5% fraud transactions in the dataset while ROC-AUC is a good metric for the imbalance data. There are 600,000 samples in the trainset and 500,000 in the testset. Since we can not get the true label of testset, we cut the last 20% samples in the trainset by datetime as the validation data, exactly same as the reality.

About 400+ original features, contains user information, shop information and transactions details and anonymous features, are provided by the official. As a supplement, 100+ features are mined from raw features in a baseline solution of the golden section. We ensure that the baseline is a top single model by the leaderboard. We run XGBoost, LightGBM, CatBoost and MT-GBM with the setting copied from the open source code.

---

[1]https://github.com/Microsoft/LightGBM
[2]https://www.kaggle.com/c/ieee-fraud-detection

## 5.3 RESULTS

Table 1: Results of models for experiment 1

| Models | best mape | mape of iteration 150 | best rmse | rmse of iteration 150 |
|---|---|---|---|---|
| XGBoost | 0.0396 | 0.0475 | 316.43 | 362.50 |
| LightGBM | 0.0438 | 0.0452 | 400.32 | 403.47 |
| NN | 0.0652 | - | 453.87 | - |
| MT-GBM | **0.0387** | **0.0423** | **308.21** | **336.82** |

In the dataset of Currency Volume, the trend of amount may have some regular patterns. Only the label of amount itself is not enough. We propose that a sub label is the volume of next day divided by current day, while the main label is the volume of next day. Two labels have different orders of magnitude while the sub label is a number floating around 1.0 and the main label is great than $10^8$.

All tree models in experiment 1 perform not badly as Table 1 shown. XGBoost is better than LightGBM at the same parameters on both metrics.

The result of first experiment shows the effects of ensemble numerical labels with different orders of magnitude. We compare the LightGBM and MT-GBM as the improvement of multi-task model, as we add the multi-task algorithm on the LightGBM repository. From Table 1, the best mape is reduced from 4.3% to 3.9% and the best rmse is reduced from 400.32 to 308.21.

We conclude the results that multi-task teach model not only the volume of currency but also the rise and fall which make model robust at small dataset by splitting at a better position, meanwhile slowing down convergence speed of the model.

Table 2: Details of sub tasks for experiment 2

| Task index | Definition | Correlation with main task |
|---|---|---|
| 1 | whether this card is fraud with same amounts in two days | 0.418 |
| 2 | whether this ip is fraud in two days | 0.309 |
| 3 | whether this card is fraud in same shop | 0.415 |
| 4 | what percentage of the transactions of same amounts is fraud in two days | 0.461 |

Table 3: Results of models for experiment 2

| Models | sub task list | 3-folds' ROC-AUC | fold 1 | fold 2 | fold 3 |
|---|---|---|---|---|---|
| XGBoost | - | 0.9443 | 0.9393 | 0.9417 | 0.9364 |
| CatBoost | - | 0.9371 | 0.9302 | 0.9309 | 0.9311 |
| LightGBM | - | 0.9442 | 0.9383 | 0.9408 | 0.9369 |
| NN-4 Tasks | [1,2,3] | 0.8410 | 0.8235 | 0.8346 | 0.8273 |
| MT-GBM-2 Tasks | [1] | 0.9447 | 0.9402 | 0.9410 | 0.9368 |
| MT-GBM-2 Tasks | [2] | 0.9441 | 0.9384 | 0.9402 | 0.9374 |
| MT-GBM-2 Tasks | [3] | 0.9446 | 0.9397 | 0.9405 | 0.9380 |
| MT-GBM-3 Tasks | [1,2] | 0.9441 | 0.9391 | 0.9388 | 0.9370 |
| MT-GBM-3 Tasks | [2,3] | 0.9448 | 0.9397 | 0.9405 | 0.9374 |
| MT-GBM-3 Tasks | [1,3] | 0.9441 | 0.9395 | 0.9396 | 0.9359 |
| MT-GBM-4 Tasks | [1,2,3] | **0.9454** | 0.9402 | 0.9407 | 0.9383 |
| MT-GBM-5 Tasks | [1,2,3,4] | 0.9443 | 0.9393 | 0.9400 | 0.9372 |

However, the second dataset has not provided multi labels too. In the risk management, fraud transaction is often divided into many classes by the type of fraud. Table 2 shows the details of sub tasks.

The dataset is divided into 3 folds in order to reduce the influence of randomness. The final result contains the average of 3 folds and single fold versions. Since we have 4 sub tasks in the case, some combinations have been tested. From Table 3, most of the multi-task experiments behave better than a single task on the ROC-AUC of the main task. We can find that 2-tasks models with sub task 1 or 3 have presentable results and the 4-tasks model has the best performance through comparative analyses between LightGBM and MT-GBM. Low correlation task means irrelevant task which is helpless to the main task and high correlation seem to have less supplement for the main task, so the task 1 and 3 are the best for this case. And the Catboost and XGBoost with single task can't reach the metric of MT-GBM. It is worthwhile mentioning that this experiment is training as the outstanding parameter $\gamma$ equals 50, cause the correlation between sub task and main task seems not so strong. And the improvement will be descended when $\gamma$ equals 10. We regard this case as an example that our algorithm makes a balance on the conflicting tasks.

Somehow the more tasks do not bring the better result. We concluded it the ensemble method of Gradients and Hessians is not perfect enough, failing to express too many tasks.

### 5.4 COMPARISON WITH NEURAL NETWORKS

We also try neural networks because it is a natural multi-task model. There are lots of big numerical numbers in the features disturbing the parameters in the neural cells. Therefore we have to transform them through $F(x)$:

$$F(x) = log_{10}(x + 1) \tag{5}$$

On the other hand, we fill the NaN value in the features with 0. And after this data preprocessing, we create a simple Multi Layer Perceptron network structure with 2 hidden layers and GELUs activation function (Hendrycks & Gimpel, 2016) to fit the dataset. The result is shown in Tables 1 and 3.

The neural network performs badly in both experiments as we explain before. It is not an unexpected result while most of these problems are solved with GBDT in the industry.

## 6 CONCLUSION

In this paper, we have proposed a new general algorithm, multi-task gradient boosted tree, basing on gradient boosted decision tree, which provides powerful features to the tree model. We have performed both theoretical analysis and experimental studies on the implementation. The experimental results are consistent with the theory and show that with the help of multi-task, MT-GBM can significantly outperform traditional GDBTs. And the MT-GBM has great extensibility as it didn't place restrictions on the datasets and the type of task.

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

## A    IMPLEMENTATION DETAILS

**Codebase**. For the Ieee-Fraud dataset, we use the codebase from [3]. The repository has done feature engineering work for the dataset from a realistic scene. We use it directly. We use the cross-validation evaluation method and divide the dataset into four folds as 3-folds train and 1 hold-out validation. For each fold, we use a hold-out set as the validation set and train a machine to learn the model from scratch using the remaining data. We set the same random seed so that the result of each fold is comparable.

**Custom Object Function**. The $G_e$, $G_u$, $H_e$, and $H_u$ can be implied in the $fobj$ function in the Train API. We build the Multi-Task framework for GBDTs and leave the custom function because we believe the high-level user would find better ensemble methods for Gradients and Hessians in different scenes.

**Hyperparameters**. For the Currency dataset, we tuning the best hyperparameters for all models. We run XGBoost, LightGBM, and MT-GBM with the setting: a small tree depth as 6, learning rate as 0.03, and lambda $L1$ as 0.1 since time-series cases is easy to overfit.

For the Ieee-Fraud dataset, we copy the hyperparameters from Github: max depth as 16, learning rate as 0.03, lambda $L1$ as 0.1, and max leaves as 500. Settings of XGBoost and Catboost have small differences to get better results.

**Comparison with neural networks**. Note that the neural network in our paper is the MLP, since other networks did not fit for tabular data. The network hyperparameters are the best in the two datasets while we get them by grip search.

## B    ADDITIONAL EXPERIMENT RESULTS

For clearness, we only show the final metric in the main text. There are some visualizations of intermediate process here.

Figures 3 and 4 show the metric during training in the Currency dataset. The lines of single task(blue and yellow) are declined fast and overfit at about iteration 150. The lines of Multi-Task(green and red) behaves better at both metrics. However, the valid metric of Multi-Task overfits finally too, as a result of a normal phenomenon in the tree model.

In Ieee-Fraud experiment we have defined 4 sub tasks for the main task. The first three of them are the different types of fraud extracted from the main task. Take the label of main task and first three tasks as $y_{gt1}$ to $y_{gt4}$, the predictions of single task LGB model as $y_{single}$, and the predictions of MT-GBM as $y_{mt1}$ to $y_{mt4}$. Figure 5 shows the Roc-auc curves of different outputs.

---

[3]https://www.kaggle.com/whitebird/ieee-internal-blend

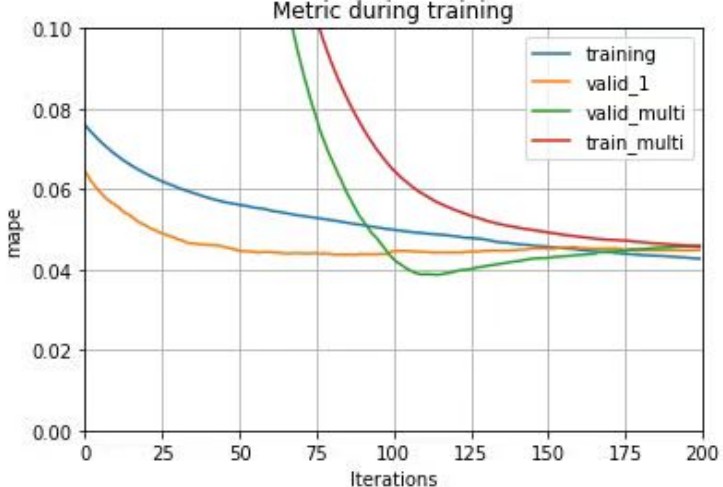

Figure 3: Train and Valid MAPE.

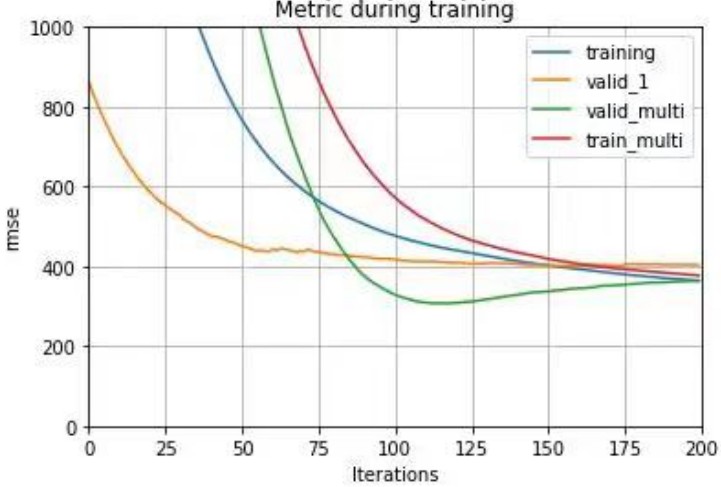

Figure 4: Train and Valid MSE.

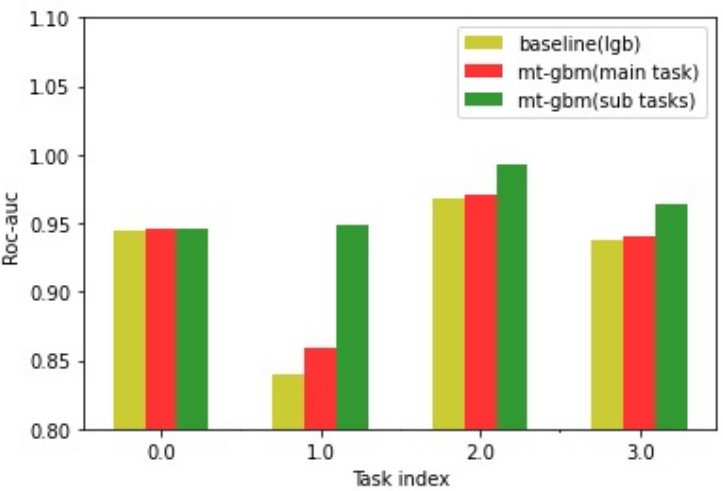

Figure 5: Roc-auc curves.

