# OpenReview forum: "MT-GBM: A Multi-Task Gradient Boosting Machine  with Shared Decision Trees"
_ICLR.cc/2022/Conference — ICLR 2022 Submitted_

### Official Review · Reviewer_UURx · 2021-10-31

**Correctness:** 1
**Technical Novelty And Significance:** 2
**Empirical Novelty And Significance:** 2
**Recommendation:** 3
**Confidence:** 4

**Main Review:**

**Strong points**
- Authors develop the algorithm for multi-task learning for gradient boosting which supports multiple tasks of different nature (classification, regression)
- Algorithm is using special gradients and hessians aggregation to define the splitting criteria which is fast and does not need large search space and recomputing every task loss in every node
- Accounting correlation between tasks to adjust learning rate during training
- Fair experiment setting: taking top algorithms on particular datasets (with necessary data processing and hyperparameters) and simply applying proposed modifications on top with demonstrating improved results

**Weak points**

*Technical correctness*
  - There is a strict motivation how loss Eq. (3) is obtained as it is optimal value when optimal weights are assigned in the leaves for a fixed tree structure. Then greedy procedure is used to construct the tree (as tree construction is NP-hard problem) based on this loss Eq. (3) and weights are assigned in the optimal way. Do authors can explain and give more details how Eq. (4) was obtained and is there any connection between weights in each leaf and loss Eq. (4)? Otherwise the provided Algorithm 2 looks artificially.
  - Algorithm 2 looks ambiguous and it is not clear why it is correct.  First, is $w_i$ defined independently for each sample or is it defined that $w_i * G_i^m$ is coming from the same normal distribution for every $m$ where m is $m$-th sample? If we define latter - it is impossible to do (by doing math one can check this). If we define independently from sample then $w_i$ and $v_i$ are normal variables which mean and std depend on the $G_i$ and $H_i$. In this case both $G_e$ and $H_e$ will be normally distributed variables which mean and std do not depend on $G_i$ and $H_i$ at all (e.g. $G_e$ is coming from sum of the same normal distributions $N(0.05, 0.01)$). The latter leads to random splitting criteria for the tree construction which is different from what authors state as contribution.
  - Algorithm 3 does not make sense to me: it is doing global scaling of optimal weights set in the leaves. Why do we need this scaling if optimal weights are assigned already in the leaves? There is no any empirical evaluation in the paper why scaling with Algorithm 3 is needed or helpful. Moreover, in the text authors say that "Therefore set the corr as a constant equal 1.0 might be better on some complex datasets which means the updating rate is same as the traditional single task GBDT" and "Trees get inconsistent paces if we set different rates for different tasks, which leads to bad results.". Thus, I think, the stated contribution "We propose a new learning algorithm that adaptively adjusts the learning rate for each task by taking the scale of residual errors into consideration. It stabilizes training on arbitrarily complex problems." is not supported by theory or experiments and could be even wrong.

*Empirical correctness*
- Absence of baseline from (Chapelle et al., 2011)
- There is no any details and ablations on how and why $corr$, $w_i$, $v_i$ are defined in Algorithms 2 and 3
- Comparison with neural networks is artificial as neural network baseline is far behind the boosting baselines, so there is no any sense to compare with it even in multi-task training
- There is lack of significance testing for the first dataset
- No any experiments with several main tasks so that they are not sub-tasks of each other

**Additional questions**
- Why not simply consider loss as a weighted sum (weights could be estimated from the task correlations on original labels) of tasks losses, compute its gradient and hessian and use Eq. (3) to construct the tree but still use multi-label assigning in the leaves (each task loss is considered here separately)? This should be considered as one of the multi-task baselines
- What is the performance of all models on the sub-tasks? This analysis would be helpful to understand that in multi-task training we improve all sub-tasks too
- Ideally there should be implementation of MT-GBM on top of XGBoost and Catboost to demonstrate that for several different boosting algorithms proposed multi-task training improves one-task training

**Clarity**

The paper needs additional work on fixing English language (a lot of sentences lacks the correct verb forms which makes hard to read and understand meaning of important sentences). Some comments on text improvements please find below:
- Related works, first paragraph: rewrite the paragraph so that it is readable
- page 3, last paragraph: rewrite the paragraph so that it is readable
- notation on supervised data: would be better to reflect the labels coming from different tasks), e.g. introduce $(x_i, y_i^1, .. y_i^n)$, thus it is clear what data and labels are given
- Eq. (1), (2) - index $t$ is absent in the right hand, but should be there
- Algorithm 1: i) $n$ should be the number of classes, not tasks (I found some ambiguity in multi-class and multi-task terms usage in Sec. 2 and in Algorithm 1); ii)  row with index 2 does not make sense if $p_{i,t}$ is probability (as it should be); why do we do exponentiation and normalization again? iii) does groundtruth label is vector with zeros and only one "1" in case we consider multi-class classification problem?
- Eq. (3) $T$ is not defined
- I don't understand this statement "while realistic datasets above always have dense outputs and multi-class which can be positive at the same time"
- "H equals constant" -> "H is constant"
- In Sec. 4 I think it is good to recap how Eq. (3) is obtained for gradient boosting
- Algorithm 2 uses $L$ but should use $L_i$, also $dp$ -> $dp_i$; there is no clear usage of $w$ and $\text{w}$,  $v$ and $\text{v}$
- Algorithm 3 uses $L$ but should use $L_i$, also $dp$ -> $dp_i$; How is the $corr$ computed? What is function $f$?
- There should be exact formula how authors assign weights/values in each leaf for multi-labels to be clear on this. Overall, would be helpful for any reader to have a quick overview of XGBoost / gradient boosting in the introduction with necessary formulas including Eq. (3) and then demonstrate all new changes introduced by authors by referring to the original gradient boosting formulas
- "When optimizing end-to-end neural networks, it is natural to sum up the gradients from all losses. However, for tree-based models, the splits that have maximal information gain for the losses can differ a lot, especially when training classification and regression tasks at the same time." I do not agree with statement on neural networks as depending on loss magnitude of its gradient can be different and we will be learning only one task. Often special loss weighting or even tricks for optimization (lr, special scheduling, clipping) are needed. I would say problems for both neural networks and GBDT, but the source/solution of the problems is different as models structure is different.
- "It seems to be a bit complex and obsolete and is beaten by the existing implementations of gradient boosted decision trees such as XGBoost and LightGBM." Do authors mean that if we do splitting criteria to optimize sum of all tasks losses (which is more computations than MT-GBM approach) on top of XGBoost then vanilla XGBoost for one main task wins?
- From presentation perspective it would be great to have (Chapelle et al., 2011) as a baseline to demonstrate how MT-GBM faster in terms of training (as simpler splitting criteria) and better in final accuracy
- Could authors confirm that MT-GBM has exactly the same input data features and all gradient boosting hyper-parameters as XGBoost, so that the only difference is the usage of Algorithm 2 and 3?

**Related works**

- I think it is worth to mention https://www.diva-portal.org/smash/get/diva2:1309070/FULLTEXT01.pdf and http://www.ms.k.u-tokyo.ac.jp/sugi/2014/TreeMTL.pdf and compare with them.
- "Therefore, it cannot learn a shared latent feature representation, which is essential for multi-task learning. ... First and foremost, the core of efficient multi-task learning is to extract valuable hidden representation that shares across all tasks, which is non-trivial for tree-based ensemble learning methods like GBDT. " Could authors refer to any work to support this statement? Otherwise, more smooth formulation like "In some works ... it was shown that share latent feature representation is essential. As soon as for neural networks shared representation works great we could inherit this idea for GBDT and model it via the shared tree structure (as a tree can be seen as latent representation)." is preferable. Maybe we don't need to have shared latent representation and other possible solutions exist which can be modeled by GBDT too.


**Summary Of The Paper:**

Recent developments in deep learning and neural networks showed impressive results on structured data, like audio, text and images. However, there are a lot of tabular data, like finance, search-engine machines, recommendation systems, telecommunications, physics and astrophysics characteristics, etc. For problems and applications with these tabular data random forest and gradient boosting (over decision trees) are the mostly used algorithms (most kaggle competitions were won by them).

Mutli-task learning is very active area nowadays as solving tasks together could improve each task (less over-fitting, better representation and generalization) and moreover we could have one general model (faster inference). It is studied in the context of deep learning a lot: computational graph mostly is shared across tasks and we can natively optimize combination of loss functions correspondent to each particular task. On the other hand, for gradient boosting (over decision trees) multi-task learning is still challenging. One of the ways is to learn shared tree structure across tasks at each gradient boosting iteration. Authors are focusing on this idea: they share same tree structure and leaves. Then authors propose the splitting criteria based on multi-task losses (instead of one-task loss as done in classic gradient boosting): gradients and hessians of tasks losses are aggregated to provide the final loss which is used as splitting criteria in the greedy algorithm of tree construction. Further, authors propose a special scaling for gradients based on correlation between the tasks (computed on losses gradients) to assign multi-label in each leaf. Proposed algorithm, called MT-GBM (multi-task gradient boosting machine) is empirically estimated on 3 datasets where subtasks are selected from the main task and should improve the main task via multi-task learning. Implementation is done on top of XGBoost and with experiments it is shown that MT-GBM improves upon the main task learning.

**Summary Of The Review:**

The problem considered in the paper is important and actual as I describe in the paper summary. The idea on developing algorithm with the shared tree construction based on tasks losses makes sense and in some sense parallel to the things done in deep learning. However, there are a lot of ambiguous, not clear places (even I think some places are wrong, see the main review) in the paper that does not allow to conclude on the correctness and applicability of the proposed approach. Moreover, the paper lacks of the proper baselines and comparisons, statistical significance testing for one of the datasets, and ablations on the parameters introduced in the algorithm 2, 3 (details in the main review). For the current paper state my recommendation is to reject due to all aforementioned things.

---

### Official Review · Reviewer_8oMi · 2021-11-02

**Correctness:** 2
**Technical Novelty And Significance:** 3
**Empirical Novelty And Significance:** 2
**Recommendation:** 3
**Confidence:** 4

**Main Review:**

One of the main strengths of this paper is its motivation and focus on the problem of multi-task learning of GBDTs. This is definitely an important problem, extending the ability of GBDTs similar to linear models and neural networks.


However, one of the critical weaknesses of this paper is its lack of clarity. It is very hard to follow the precise problem being addressed in the paper and the precise choices made in the proposed scheme. This leads to various questions such as the following:

- Regarding "for tree-based models", the splits that have ... are suitable for all tasks."
  - Why is simple sum an issue when it finds a split that is "good enough" for all tasks? It works for neural networks, why should it be an issue with GBDTs especially if we consider a scaled sum of per-task losses?
  - The proposed "shared tree" is probably essentially finding a tree split "suitable" for all tasks.
- Regarding Figure 1:
  - The motivating example is hard to follow. The intuition seems reasonable but the description of the specifics is hard to understand for me. I think we need to start from describing what each task is and why same sample has two labels.
  - Is this a MT case where all tasks have the same covariates but different task-specific labels? That is, as training data, we have the same $x$ values but different $y$ values for each $x$ corresponding to each task?
  - What is a "noisy task"?
- Regarding "main task objective function $L_1$ as the metric to evaluate the model"
  - Is this a MT setting where we evaluate only on one of the tasks? This does not appear to be standard practice. Can you please provide motivation for such a MT setting?
- Regarding the positioning against Chapelle et al. (2011):
  - The issue raised by the authors appear to be somewhat non-technical -- seems like the authors are claiming that minimizing a sum of relative losses over all tasks is "complex and obsolete" compared to XGBoost & LightGBM. It would be helpful to have this elaborated in more technical terms.
- Regarding comment on GBDTs for multi-class classification while discussing Fan et al., 2017:
  - It is an unclear comment and citation -- training one tree per class is standard in multiclass classification when minimizing the multinomial loss (or the cross-entropy loss) with GBDT. It is odd to cite a 2017 paper when this has been around since early 2000s.
  - It is not clear what is meant by "activation" during "inference" based on specified class -- if we knew the "specified class" we won't need inference. This comment needs further clarification.
- Regarding "each sample in the dataset should have $n$ predictions, .... could only be calculated by one Gradient and one Hessian for each sample"
  - Is this implying the situation where all tasks have the same covariates but different labels? This is an unusual MT setting. Can you provide practical scenarios where this is true?
  - The "problem" in the histogram object needs a bit more detailed description.

It would very helpful to start with a clear definition of the problem being considered, followed by the motivating example, connecting it to the aforementioned problem definition. Given the proper problem definition and motivation, it would be easier to distinguish between existing literature on boosting in a multi-task setup. In the current form, it is not clear why the existing schemes for multi-task boosted trees are not considered as baselines.


Moreover, the proposed scheme appears to lack useful motivation and justification, making algorithms such as Algorithm 2 & 3 quite unintuitive. Specifically, I have the following concerns/questions/comments:

- In Algorithm 2, it appears that the gradients are first scaled randomly across tasks (Algorithm 2, Line 6) and then the tasks are effectively chosen randomly and their (weighted) sum is used as the gradient to use for splitting (lines 7-11). This form of scaling and task-sampling seems very unintuitive to me without proper justification. Why can we not just consider simple normalization (across tasks to make the losses comparable across tasks) followed by a simple sum for the splitting gradient? This is important to discuss in my opinion.
- There are magic numbers (0.05 for mean, 0.01 for standard deviation across tasks, [10, 100] as range for $\gamma$) without any discussion or motivation. These are essentially hyperparameters of the proposed scheme and should be dealt with as such.
- Gradient for leaf updates are also somewhat confusing in the Algorithm 3 -- it is not clear how an $n$-way correlation coefficient is computed and also how that allows for different "learning rates" for different tasks. If all the gradients for all tasks are just scaled with a single scalar, I am not sure it is doing anything. Maybe I misunderstand the procedure. Additionally, it would be good to understand how much effect this single-scalar-scaling has on the final performance (if any).


Finally, the empirical evaluation is quite limited, with just two data sets and an artificially created multi-task setup. It is not clear how well the hyperparameters of the XGBoost, LightGBM and neural network baselines were optimized (if at all). Given the randomness present in MT-GBM, and the small margin of improvement over baselines, it is also important to have confidence intervals around the performance metrics utilizing different seeds and/or different train/test splits. It would be good to focus on a more natural multi-task setup with a clear multi-task metric to compare methods. Moreover, it would be good to study the effect of the different choices made in MT-GBM -- for example, compare a simple scaling + sum of multi-task losses to the proposed random scaling + random task selected losses in Algorithm 2 (something similar can also be done with Algorithm 3). This would help us highlight the gains obtained from the choices made by the authors.



Additional comments:

Wrong citation for XGBoost (Roe et al., 2004)! The correction citation is _"Chen, Tianqi, and Carlos Guestrin. "Xgboost: A scalable tree boosting system." Proceedings of the 22nd acm sigkdd international conference on knowledge discovery and data mining. 2016."_


**Summary Of The Paper:**

This paper focuses on the multi-task learning for the popular gradient boosted decision trees (GBDTs) for tabular data. This paper proposes two novel techniques to the learning of GBDTs -- the multi-output tree and the mutli-task splitting -- that can be easily incorporated to the existing implementations of GBDTs without significant computational overhead. The multi-output tree shares the tree structure (split dimensions and thresholds) across all tasks but maintains different leaf statistics for each task. The multi-task splitting utilizes the per-task statistics in an internal node to generate a single statistic (gradient and hessian) that incorporates information from a small number of tasks to find the optimal split. The leaf value updating procedure is also modified to handle the multi-task setup. The proposed MT-GBM scheme is evaluated on two data sets where there is a single "main" tasks and auxilliary tasks are generated and used in the multi-task learning of MT-GBM. The empirical results indicate that the proposed MT-GBM is able to marginally outperform XGBoost and LightGBM trained on a single task, and a neural network trained with multiple tasks.



**Summary Of The Review:**

I recommend a rejection based on the weaknesses I list above. Specifically, in the current state, it is very hard for me to follow the proposed scheme and why existing schemes do not apply. It is important that the relevant baselines are properly compared and contrasted against. Moreover, we only see a marginal improvement of the proposed scheme over the considered baselines.

---

### Official Review · Reviewer_tQ17 · 2021-11-03

**Correctness:** 2
**Technical Novelty And Significance:** 3
**Empirical Novelty And Significance:** 2
**Recommendation:** 3
**Confidence:** 3

**Main Review:**

Strengths:
- (S1) Trying to adapt multi-task learning (MTL) approaches to GBMs is a natural and timely topic

Weaknesses:
- (W1) Comparison to related work: The comparisons to prior work like Chapelle et al 2011 and Fan et al 2017 are presented with very high-level, hand-wavey arguments for why these prior approaches are insufficient.  A combination of clearer arguments and actual comparison in the experimental section would be necessary to properly compare here.
- (W1a) Also, the authors miss some more recent works that should probably be included, e.g. http://proceedings.mlr.press/v108/han20a/han20a.pdf
- (W2) Imprecise arguments: Overall, the paper is written with many heuristic claims throughout that are not tied to specific formal or empirical justification.
- (W3) Insufficient empirical evaluation:
  * (W3a) The experiments, as noted in #W1, don't compare to prior approaches in this area, and should have done this.
  * (W3b) The experiments compare to one very simple DNN MTL model, and two single-task GBM models.  This is both overall an insufficient set of baselines (given the many very sophisticated and competitive models that could be used here), and also does not properly disentangle/ablate the effect of multi- vs. single-task from the specific advantages of the particular MTL-GBM approach considered here.
  * (W3c) Insufficient ablations of method
  * (W3d) Overall, the experiments section in particular is written in a very imprecise and unclear fashion.

**Summary Of The Paper:**

This paper proposes a new multi-task variant of Gradient Boosted Machines (GBMs), a corresponding learning algorithm that adaptively adjusts the per-task learning rate, and implements a version on the LightGBM framework to show empirical results.

**Summary Of The Review:**

Overall, the paper proposes an interesting new method, but has an insufficient empirical evaluation and is additionally quite imprecise throughout.

---

### Official Review · Reviewer_CBKT · 2021-11-05

**Correctness:** 3
**Technical Novelty And Significance:** 2
**Empirical Novelty And Significance:** 2
**Recommendation:** 3
**Confidence:** 3

**Main Review:**

Authors claim “However, it is known that neural networks sometimes fail in applications with tabular data.” in the introductions. Can you cite the relevant work here?
Rephrase “It seems to be a bit complex and obsolete and is beaten by the existing implementations of gradient boosted decision trees such as XGBoost and LightGBM.”
The related works section is very short.
Considering that boosted decision trees are not the most popular approach in ML now, the authors could ease the readers by providing more background on the topic.
The authors evaluate their approach on just 2 datasets. More evaluation is needed to make a convincing argument for their approach.


**Summary Of The Paper:**

Authors propose a way to perform multi-task learning with gradient boosted decision trees. Multitask learning has shown promise in deep learning and other machine learning systems. It is hard to perform multitask learning in decision trees. The main contributions from the paper are as follows
Authors propose a multitasking gradient boosted decision tree
Authors further propose a way to dynamically adjust the learning rate for each task so the learner does not overfit any specific task.
They evaluate their approach on two public datasets.


**Summary Of The Review:**

The authors address a real problem with gradient boosted decision trees. But the results are limited and do not appear very convincing. They need to evaluate their approach on more datasets to make a convincing argument. The improvements from the proposed approach do not look very significant either.

---

### Public Comment · ~Zhendong_Zhang1 · 2022-03-08
**Compare with "GBDT-MO: Gradient-boosted decision trees for multiple outputs"**

This paper extends GBDT for multi-task. Its key idea is that the authors construct multiple outputs in each leaf.
This idea has been explored in the paper "GBDT-MO: Gradient-boosted decision trees for multiple outputs" which is published in "IEEE transactions on neural networks and learning systems,2020".

I think the authors should discuss differences and similarities compared with GBDT-MO.

---

### Decision · Program_Chairs · 2022-01-20

**Decision:**

Reject

**Comment:**

This paper proposes a multi-task version of Gradient Boosted Machines (GBMs). The paper proposes a learning algorithm that adaptively adjusts the learning rate per task. Empirical evaluation is carried out on two datasets with the method implemented in the LightGBM framework.

The reviewers thought that the paper is not very clear. They were not ready to accept the paper claims based on the current version. In particular, the algorithms are hard to follow, the empirical evaluation is not easy to follow and there are missing comparisons to related work. The authors did not offer a response to the reviews.